# Single Train Multi Deploy on Topology Search Spaces using Kshot-Hypernet

## Abstract

Neural Architecture Search (NAS) has become a crucial research direction for automating the design of neural networks. The introduction of weight sharing has significantly reduced the computational and time costs of NAS. Recent approaches enable the simultaneous training of numerous sub-networks without the need for retraining; however, these methods are primarily limited to the Size Search Space (*SSS*), which provides limited architecture diversity. To date, one-shot training method based on the more diverse Topology Search Space (*TSS*) remains unexplored. *TSS* has greater potential for hardware-aware architecture search. In this work, we propose a novel NAS method that operates on *TSS*, while maintaining high efficiency. To do so, we introduce Kshot-Hypernet, that extends in-place distillation to TSS, significantly improving supernetwork training. Experiments on NASBench-201 show that, once the supernet is trained, most sub-networks can match or even exceed the performance of those trained from scratch. Furthermore, our method achieves 80.7% top-1 accuracy on ImageNet with only 8.7M parameters.

## 1 Introduction

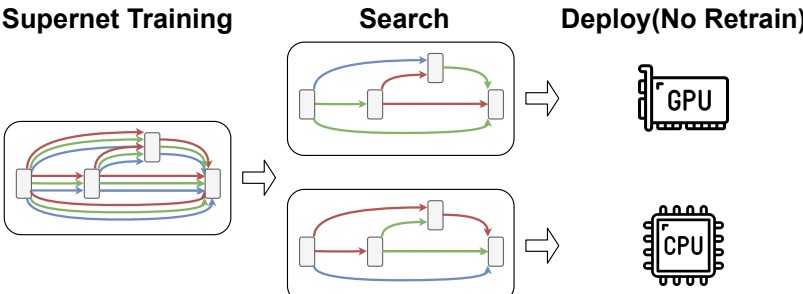

Figure 1: TSS-based Supernetwork Training and Deployment Workflow. Different colored arrows represent different operations, and the nodes denote inputs and outputs.

Neural Architecture Search (NAS) has gained attention as an alternative to manual network design, which is time-consuming and requires detailed expert knowledge. The primary goal of NAS is to automatically discover high-performing architectures for specific tasks. Early works Zoph & Le (2017); Real et al. (2019); Zoph et al. (2018) demonstrated the effectiveness of NAS, but at the cost of substantial computational resources, as they required training numerous candidate architectures from scratch for evaluation.

The introduction of weight sharing by Efficient NAS (ENAS) greatly improved the efficiency and reduced the resource consumption of NAS (Pham et al., 2018). With the increasing demand for deploying neural networks across diverse platforms and devices, NAS objectives have expanded to include the design of architectures tailored for various hardware. Due to platform-specific characteristics, the same architecture may exhibit

vastly different performance on different devices (Wu et al., 2019). Consequently, efficient deployment often necessitates repeated architecture searches for each target platform. If each discovered architecture must be trained from scratch, the computational cost becomes prohibitive.

To address this, recent works Cai et al. (2020); Yu et al. (2020a) have proposed training a single large supernetwork using weight sharing, from which all subnetworks' weights can be directly sampled and subsequently used for different platforms. This approach enables rapid validation and deployment of architectures without the need for retraining from scratch, significantly reducing the overall NAS cost.

However, most current methods for multi-deploy NAS without retrain are based on the Size Search Space *(SSS)*, where different sizes of network components are explored, such as the MobileNet search space (Howard et al., 2019). Although *SSS* can cover a large number of candidate architectures and is effective for device-aware search, it suffers from limited structural diversity and hardware adaptability. In *SSS*, subnet variations are mainly achieved by adjusting parameters such as network width (number of channels), depth (number of layers), and convolution kernel size, while the overall network structure remains fixed (e.g., all using MobileNet blocks). All candidate networks are essentially variants of existing architectures, restricting the discovery of novel structures. Consequently, performance differences among subnets are primarily determined by parameter count and FLOPS, rather than by architectural innovation.

The Topology Search Space *(TSS)* utilized by earlier NAS approaches (Zoph & Le, 2017; Real et al., 2019; Zoph et al., 2018) is different from *SSS*. Its architectures are represented as directed acyclic graphs (DAGs) composed of diverse operators.

*TSS* defines the search space by specifying the number of nodes in a DAG, the connections between nodes, and the set of candidate operations, forming so-called search blocks. Networks are constructed by stacking these search blocks, which can either share the same architecture or be individually designed. Unlike *SSS*, *TSS* allows for substantial architectural diversity among subnets, as the operations within each search block can be entirely different. This flexibility enables the discovery of architectures better suited to specific hardware accelerators, making NAS more meaningful for hardware-aware optimization. However, the increased diversity and complexity of *TSS* also make supernetwork training significantly more challenging. As a result, most existing *TSS*-based works (e.g. Su et al. (2021); Zhao et al. (2021); Hu et al. (2020)) focus on improving the ranking of subnets within the supernetwork, but still require retraining the discovered architectures from scratch to achieve high performance.

In this work, we tackle the persistent challenge that topology search space *TSS*-based NAS methods typically require retraining to achieve deployable performance. To address this limitation, we propose a novel supernetwork training framework that integrates multiple advanced techniques. Our focus is on enabling subnets discovered through a single training of the supernetwork to achieve deployable accuracy without retraining, rather than on hardware adaptability. The main contributions of this paper are as follows:

- We introduce **Kshot-Hypernet**, an enhanced version of Hypernet (Su et al., 2021), which incorporates KshotNAS (Zhao et al., 2021) to improve the capacity of supernetworks in *TSS*.

- We develop **Focus-Fair Sampling** and a customized distillation strategy to facilitate more effective supernetwork training.

- We achieve competitive results on NAS-Bench-201 and ImageNet, enabling direct deployment without retraining.

For NAS-Bench-201 (Dong & Yang, 2020), our approach achieves an average accuracy of 87.12% across all subnets (compared to 87.06% when trained from scratch), with the best subnet reaching 92.47% (vs. 94.37% from scratch) on CIFAR-10. On CIFAR-100, the average accuracy is 61.03% (vs. 61.41%), and the best is 71.98% (vs. 73.51%). On ImageNet, our method attains a top-1 accuracy of 80.7% with only 8.7M parameters, matching the SOTA performance.

## 2    Related Work

The enormous computational resources and time consumption required by conventional NAS (Zoph & Le, 2017; Real et al., 2019; Zoph et al., 2018; Liu et al., 2018) hinders the widespread adoption of NAS. The introduction of weight sharing (Pham et al., 2018), where a supernetwork encompassing all architectures in the search space is trained only once and subnetworks are obtained by sampling its weights during searching, greatly reduces both computational and time costs up to $1000\times$ compared to conventional NAS. Differentiable Architecture Search (DARTS) (Liu et al., 2019) assigns coefficients to each path, making the output of each node a weighted sum of all paths. This technique enables differentiable search, allowing optimal architectures to be found via gradient descent. ProxylessNAS (Cai et al., 2019) introduces additional architectural parameters during training and uses binary encoding to activate only one path at a time, further improving search efficiency.

### 2.1    Subnetwork Sampling

The subnetwork sampling strategy plays a crucial role in determining the final performance of the supernetwork. OneShot-NAS (Bender et al., 2018) adopts path dropout during training, with the dropout rate increasing over time. Single Path One-Shot (SPOS) (Guo et al., 2020) compresses the search space so that all subnetworks are single-path, randomly selecting one path per iteration, thus treating the supernetwork as a framework rather than fully training it. FairNAS (Chu et al., 2021) introduces a fair sampling strategy and aggregates gradients for simultaneous updates, addressing the subnetwork iteration order issue in SPOS and reducing the optimization gap between subnetworks. DFairNAS (Meng & Chen, 2023) further improves upon FairNAS by scoring all operations based on subnetwork performance, encouraging the combination of high-scoring operations. Inspired by these works, we propose a novel FocusFair sampling method, which increases the probability of sampling high-performing subnetworks while minimizing the impact on the remaining subnetworks.

### 2.2    Size Search Space based NAS

Single-Path NAS (Stamoulis et al., 2019) utilizes the MobileNet search space and introduces convolutional kernel sharing, where smaller kernels inherit properties from larger ones. MobileNetV3 (Howard et al., 2019) applies NAS to search for networks of various sizes, setting a foundation for subsequent *SSS*-based NAS methods. Once-for-All (OFA) (Cai et al., 2020) employs progressive shrinkage and fine-tuning on a fully trained supernetwork, enabling direct deployment of subnetworks without retraining. BigNAS (Yu et al., 2020a) incorporates several training techniques, such as the sandwich rule (Yu & Huang, 2019), in-place distillation (Yu & Huang, 2019), and exponentially decaying with constant ending, to achieve results comparable to OFA.

However, as outlined above, these methods are based on a limited search space and do not explore truely novel architectures; all discovered networks are essentially MobileNet variants. Nevertheless, the techniques proposed, such as in-place distillation and the transformation matrix for convolutional kernels in OFA, are highly valuable. We extend in-place distillation to *TSS*, significantly improving supernetwork training. The transformation matrix aims to prevent complete weight sharing among subnetworks, enhancing the supernetwork's representation capacity, which aligns with our use of Hypernetwork. In addition, Autoformer (Chen et al., 2021) introduces the concept of weight entanglement and extends it from CNNs to transformers. Other related works that enable deployment without retraining include Hardware-Aware Transformers (HAT) (Wang et al., 2020), Focusformer (Liu et al., 2022b), AttentiveNAS (Wang et al., 2021), NASVIT (Gong et al., 2022), and ShiftNAS (Zhang et al., 2023).

### 2.3    Rank Correlation

The effectiveness of weight sharing in NAS has not been theoretically proven and relies on the assumption that the ranking of subnetworks evaluated using the supernetwork is consistent with that obtained by training each subnetwork from scratch. Many studies have focused on verifying or improving the correlation between these two rankings. For example, Hu et al. (2020) proposed an angle-based method to shrink the search

space and enhance ranking correlation. Zhang et al. (2020b) demonstrated that ranking correlation based on weight sharing can be unstable due to interference among subnetworks. Their research on group sharing indicates that grouping subnetworks by architectural similarity can reduce the number of subnetworks while improving ranking correlation. FewShot-NAS (Zhao et al., 2021) extended OneShot-NAS to use multiple supernetworks, showing that increasing the number of supernetworks leads to better ranking correlation, consistent with the findings of Zhang et al. (2020b). Liu et al. (2022a) further improved FewShot-NAS by gradually increasing the number of groups. KShot-NAS (Su et al., 2021) assigns $K$ weights to each convolutional layer and uses a simplexnet to encode the architecture and output $K$ weight coefficients, which are then combined to form the final network weights. In contrast, our method enables all subnetworks to be trained to a directly deployable state, making ranking correlation less critical.

### 2.4 Hypernetwork based NAS

Two notable works (Brock et al., 2017; Zhang et al., 2020a) employ Hypernetworks (Ha et al., 2016) in a manner similar to ours to generate network weights. SMASH (Brock et al., 2017) encodes the network architecture as a 3D tensor and uses a 26-layer DenseNet (Huang et al., 2018) to generate all network weights in a single forward pass. Graph HyperNetworks (GHN) (Zhang et al., 2020a) represent the network architecture as a computational graph to generate weights. The key advantage of Hypernetworks in NAS is their ability to generate weights conditioned on the architecture encoding, thus avoiding complete weight sharing and potentially improving the achievable performance of subnetworks.

## 3 Method

Unlike the previous methods without retraining, we make it possible to perform multiple searches and deployments after training the supernetwork based on *TSS* once. Our method consists of two main parts:

- We adapt Hypernetwork for NAS and integrate it with KshotNAS, enhancing the expressiveness of Hypernetwork. This allows the weights of architectures in the search space to be less shared, thus achieving higher architecture diversity.

- We propose a novel supernetwork training process based on the *TSS*, incorporating distillation and Focus-Fair sampling methods.

We first describe the combination of Hypernetwork and KshotNAS in Section 3.1, followed by the distillation and sampling method, employed in Kshot-Hypernet.

### 3.1 Kshot-Hypernet

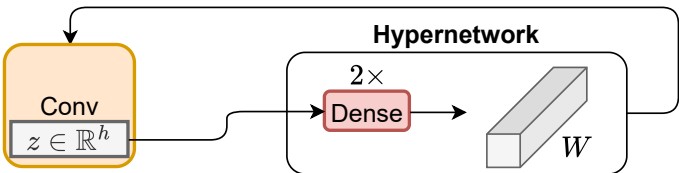

Figure 2: Hypernetwork as proposed in Ha et al. (2016). The convolutional layer receives a feature vector $z \in \mathbb{R}^h$ as input, which is utilized by the hypernetwork to generate the corresponding convolutional weights.

Kshot-Hypernet combines Hypernetwork (Ha et al., 2016) with KshotNAS (Su et al., 2021). The core concept of Hypernetwork is weight decomposition: the weights of convolutional layers are generated by a shared Hypernetwork, which significantly reduces the number of parameters. Specifically, consider a convolutional neural network with $d$ layers. The weight of the $i$-th layer, $W^i \in \mathbb{R}^{C_{in}^i \times C_{out}^i \times K^i \times K^i}$, where $C_{in}^i$ and $C_{out}^i$

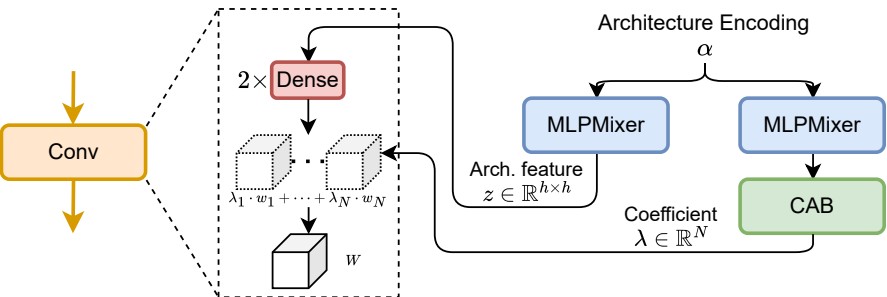

Figure 3: Kshot-Hypernet. The convolutional layer is implemented with two fully connected layers, which take both architecture features and coefficients as input to generate the convolutional weights.

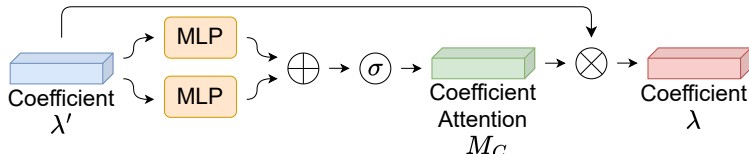

Figure 4: Coefficient Attention Block.

denote the input and output channels and $K^i$ is the kernel size, is generated as:

$$W^i = g(z^i) \, , \tag{1}$$

where $z^i \in \mathbb{R}^h$ is a learnable feature vector of size $h$ for each layer, and $g$ is a generative function implemented by two fully connected layers (see Figure 2). The weight dimensions of these layers are $h \times (C_{in}^i \cdot h)$ and $h \times (C_{out}^i \cdot K^i \cdot K^i)$, respectively. Since the input and output channels of each convolutional layer may differ, the Hypernetwork output is fixed to a unit convolution kernel, such as $16 \times 16 \times 3 \times 3$, and multiple unit kernels are stacked to construct the final weights.

Although the original Hypernetwork significantly reduces the number of parameters, its impact on network performance remains non-negligible. To better balance parameter efficiency and model performance, we assign each convolutional layer an independent weight generation network. While the original method shares the weight generation network and uses independent feature vectors for each unit convolution kernel, our approach adopts layer-specific weight generation networks but shares the feature vectors across layers. This design is partially inspired by Li et al. (2021), though in their method, feature vectors are derived from the previous layer's output and are not shared. In our framework, feature vectors are generated from the architecture encoding using a dedicated lightweight feedforward network, which is more suitable for NAS. This feedforward network consists of several MLPMixer layers (Tolstikhin et al., 2021). To further reduce parameter count, we increase the size of the shared component (architecture features) and decrease the size of the independent component (weight generation network). The resulting weights are given by $W^i = g^i(z)$, analogous to equation 1, but where $z \in \mathbb{R}^{h \times h}$ is the feature vector generated by the architecture encoding, and the weight dimensions of the two fully connected layers of $g^i$ are $h \times C_{in}^i$ and $h \times C_{out}^i \cdot K^i \cdot K^i$, respectively. However, even with these improvements, the expressive power of the Hypernetwork remains limited. To further enhance its capacity, we integrate the KshotNAS approach with weight decomposition.

### 3.2 Weight decomposition for KshotNAS

In KshotNAS, increasing the number of weights $N$ in the weight dictionary leads to a rapid growth in parameters, making network training challenging (Su et al., 2021). A key advantage of Hypernetwork is its parameter efficiency. By combining both methods, we can employ a larger $N$ to improve expressiveness

without significantly increasing the training complexity. In our approach, the parameter increment mainly comes from the first fully connected layer of the weight generation network, i.e., $h \times C_{in}^i \cdot N$. The weight dictionary is defined as:

$$\Theta_{W^i} = [w_1^i, ..., w_N^i] = g^i(z) \ , \tag{2}$$

where $w_n^i \in \mathbb{R}^{C_{out}^i \times C_{in}^i \times K^i \times K^i}$. Assume a convolutional layer with input and output channels $C_{in}$ and $C_{out}$ both set to 256, kernel size $K = 3$, hidden dimension $h = 64$, and number of generated weights $N = 64$. Under these settings, the original method requires $C_{out} \cdot C_{out} \cdot K^2 \cdot N = 37.75M$ parameters, whereas our method only needs $C_{in} \cdot h \cdot N + C_{out} \cdot K^2 = 1.05M$, reducing the parameter count by nearly $35\times$. Additionally, K-shot employs SimplexNet to generate weight coefficients, which is conceptually similar to the shared component in our network. By combining these approaches, we simultaneously generate both weight coefficients and architecture features. This enables the generation of distinct weight dictionaries and corresponding coefficients for different architectures, significantly enhancing the expressiveness of the weight dictionary. The final weights are computed as:

$$W^i = \sum_{n=1}^{N} \lambda_n w_n^i \ , \tag{3}$$

$$\lambda = softmax(f(\alpha)) \ , \tag{4}$$

where $\lambda \in \mathbb{R}^N$ denotes the weight coefficients, $\alpha$ is the architecture encoding, and $f$ is a multi-layer MLP-Mixer. To further boost network performance, we insert a modified channel attention module (CAM) (Woo et al., 2018) before the softmax operation, which was originally proposed for self-attention on the channel dimension in convolutional neural networks. We refer to this as the Coefficient Attention Block (CAB) (see Figure 4). We use $f(\alpha)$ as the input to CAB to enhance important coefficients. We also experimented with traditional attention mechanisms, but found that CAB performed better.

### 3.3 Supernetwork Training on Topology Search Spaces

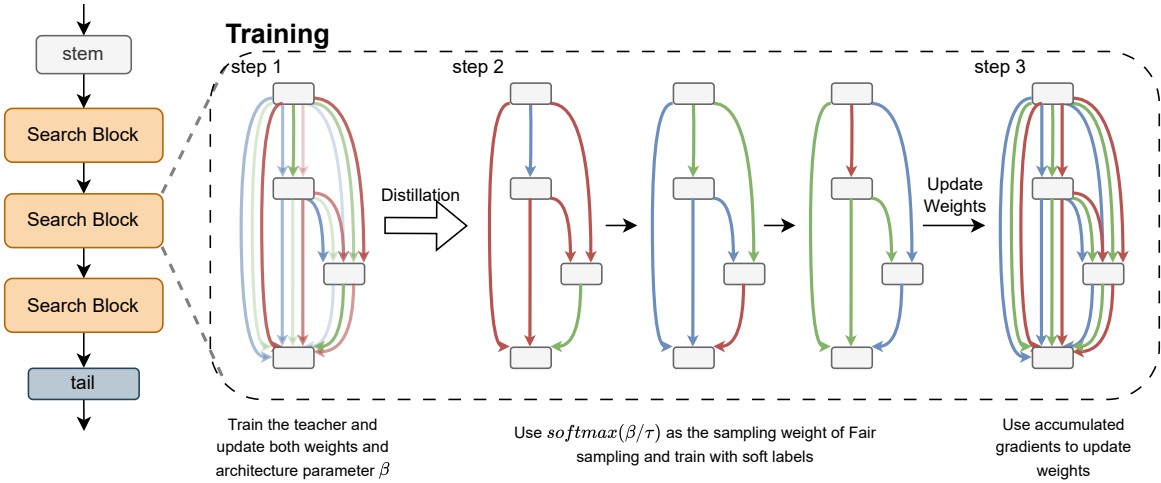

Figure 5: Training flowchart. First, the complete supernetwork with architecture parameters is trained, including both forward and backward passes (the color intensity indicates the magnitude of the architecture parameters). Next, subnetworks are sampled using the Focus-Fair sampling method based on the architecture parameters, and are distilled using soft labels obtained from the first step. Finally, the accumulated gradients are used to update the supernetwork weights. The weights of all convolutional layers in the supernetwork are generated by Kshot-Hypernet.

Training a supernetwork based on *TSS* is more challenging than training one based on *SSS*. In *SSS*, such as the MobileNet search space, the smallest subnetwork is the shared part of all networks; thus, all subnetworks can be viewed as pruned versions of the supernetwork. From this perspective, the original weight-sharing NAS process can be improved by omitting the retraining step and adopting progressive shrinking, consistent with pruning (Cai et al., 2020). A more aggressive strategy is to train all architectures simultaneously without any fine-tuning (Yu et al., 2020a). However, in *TSS*, architectures do not have a hierarchical relationship and only partially overlap, making it much harder to train all subnetworks at once. To address these challenges, we propose novel sampling and distillation strategies for supernetwork training, as illustrated in Figure 5. Our Kshot-Hypernet generates the weights of convolutional layers during supernetwork training, i.e., training the supernetwork is essentially the process of training the Kshot-Hypernet.

**Distillation in topology search space:** Knowledge distillation is an effective technique for improving network performance. BigNAS (Yu et al., 2020a) demonstrated that using the largest subnetwork for in-place distillation (Yu & Huang, 2019) is effective in *SSS*. However, in *TSS*-based NAS, the architectural differences between subnetworks make uniform distillation challenging. Although larger networks typically yield better performance, in *TSS*, the largest subnetwork does not inherently subsume the others, and thus may not be the best teacher for distillation. Instead, the entire supernetwork, which contains all subnetworks and their characteristics, serves as a more suitable teacher. Inspired by DARTS (Liu et al., 2019), we introduce additional architecture parameters $\beta$ to balance the outputs of different operations:

$$x^j = \sum_{o \in \mathcal{O}^{i,j}} \frac{\exp(\beta_o^{i,j})}{\sum\limits_{o' \in \mathcal{O}^{i,j}} \exp(\beta_{o'}^{i,j})} o^{i,j}(x^i) \,, \tag{5}$$

where $\mathcal{O}^{i,j}$ denotes the set of all operations between node $x^i$ and node $x^j$. This approach, which aggregates all operations, makes the supernetwork a better teacher for all subnetworks than any single-path network. Unlike DARTS, which aims to search for the best architecture on the validation set, our goal is to train an optimal teacher for all subnetworks.

Additionally, we do not freeze the BatchNorm parameters as in DARTS, even though this may affect the scaling of $\beta$. This is because: 1) we train the supernetwork and subnetworks jointly, so BatchNorm parameters benefit subnetwork training and can be corrected during subnetwork updates; 2) our objective is to train the best teacher for all subnetworks, not to search for the best architecture, so the scaling of $\beta$ is less critical.

As illustrated in Figure 5, for each training batch, we first use the teacher (supernetwork) to generate soft labels and update both the network weights and architecture parameters $\beta$. Then, subnetworks are sampled and trained on the same data, using only the soft labels for loss computation. We have experimented with combining distillation and target losses, but found that using only soft labels yields better results.

**FocusFair Sampling:** The sampling strategy plays a crucial role in the training of weight-sharing NAS. As noted in FairNAS (Chu et al., 2021), uniform sampling can introduce sequence bias, which FairNAS partially alleviates. Yu et al. (2020b) further showed that FairNAS offers better stability than uniform sampling, especially in the early training stages. Moreover, as illustrated in Figure 6a, our experiments reveal that, in *TSS*, low-performance architectures tend to benefit more from supernetwork training, often improving to match or surpass their from-scratch performance. Conversely, high-performance architectures are more likely to be negatively impacted, resulting in performance significantly below their from-scratch counterparts.

To address this, we propose Focus-Fair Sampling, which focuses training on high-performance architectures while minimizing adverse effects on others. Inspired by DARTS (Liu et al., 2019), we use $softmax(\beta)$ as the sampling weight in FairNAS, so that operations with higher architecture parameters $\beta$ are more likely to be selected together, while still ensuring all operations are traversed in each iteration. This approach minimizes the impact on non-high-performance architectures.

For example, consider a search block with three nodes $[x^0, x^1, x^2]$ and operation pool $[o_0, o_1, o_2]$, with $softmax(\beta^{0,1}) = [0.2, 0.5, 0.3]$ and $softmax(\beta^{1,2}) = [0.3, 0.1, 0.6]$. The probability of sampling $[o_1^{0,1}, o_2^{1,2}]$ in the first iteration is 30%, nearly twice that of uniform sampling (11.1%). Combined with our distillation

method, this sampling is cost-free. Compared to DFair (Meng & Chen, 2023), our method is simpler and does not require extensive validation during training.

To prevent $max(softmax(\beta))$ from becoming too large—i.e., over-focusing on a single high-performance architecture and causing training imbalance—we introduce a temperature hyperparameter $\tau$, using $softmax(\beta/\tau)$ as the sampling weight to smooth the distribution. Empirically, we find $\tau \approx 1.5$ yields the best results.

## 4 Result

In this section, we present the experimental results of our method on NAS-Bench-201 (Dong & Yang, 2020) and ImageNet-1K (Russakovsky et al., 2015).

### 4.1 Evaluation on NAS-Bench-201

To evaluate the overall performance of our method across the entire search space, we use NAS-Bench-201 (Dong & Yang, 2020) for testing. NAS-Bench-201 is a NAS benchmark based on cell search, where the search space is defined by DAG. Each cell contains four nodes, and five possible operations can be selected between any two nodes: zeroing, skip connection, $1 \times 1$ convolution, $3 \times 3$ convolution, and $3 \times 3$ average pooling. This results in a total of 15,625 possible architectures. For each architecture, NAS-Bench-201 provides detailed training data for both 12 and 200 epochs on three datasets: CIFAR-10, CIFAR-100, and ImageNet16-120. As such, NAS-Bench-201 is widely used to evaluate both the search capability of NAS methods and the ranking ability of weight-sharing approaches. We assessed the training effectiveness of the supernetwork, specifically whether subnetworks sampled from the supernetwork can directly achieve the same performance as training from scratch. Therefore, we use the average accuracy of all subnetworks as our evaluation metric.

**Supernetwork Training:** For the Hypernetwork configuration, we set $h = 32$ and $N = 64$. Training is performed using stochastic gradient descent (SGD) with a momentum of 0.9 and Nesterov acceleration, with a batch size of 256 per GPU. The initial learning rate is set to 0.2, and the total training duration is 300 epochs. Training details are provided in Appendix A.1.

Table 1: Average accuracy on NAS-Bench-201.

|  | Cifar10 | Cifar100 | Imagenet16 |
|---|---|---|---|
| Baseline | 87.06% | 61.41% | 33.59% |
| DARTS Liu et al. (2019) | 11.59% | 1.61% | 0.97% |
| FairNAS Chu et al. (2021) | 77.46% | 41.19% | 18.57% |
| Ours | 87.12% | 61.02% | 29.79% |

Table 2: Accuracy of best model on NAS-Bench-201.

|  | Cifar10 | Cifar100 | Imagenet16 |
|---|---|---|---|
| Baseline | 94.37% | 73.51% | 47.31% |
| DARTS Liu et al. (2019) | 40.79% | 8.12% | 2.97% |
| FairNAS Chu et al. (2021) | 86.61% | 52.00% | 25.50% |
| Ours | 92.47% | 72.04% | 46.33% |

**Result:** Tables 1 and 2 present the training results of our method on NAS-Bench-201. The official training from scratch results supplied by the benchmark serve as the baseline. All reported accuracies are measured using weights directly sampled from the supernetwork, without any additional retraining or fine-tuning.

Our method achieves a slightly higher average accuracy on CIFAR-10 compared to the baseline, and is very close to the baseline on CIFAR-100. The accuracy difference of the best model between our method and the baseline is within 2%. On ImageNet16-120, the accuracy gap is larger, which may be due to the dataset being more challenging to train and lacking suitable training techniques. DARTS (Liu et al., 2019), designed primarily for the search (optimization) phase, does not yield well-trained standalone subnetworks. FairNAS (Chu et al., 2021) emphasizes sampling balance, while our method significantly improves both the best model and average accuracy. Overall, our method outperforms previous training methods on NAS-Bench-201 in training all subnetworks. Although there is still a gap in the best model, when the search space is sufficiently large, constraints are typically imposed during the search phase to limit model size, and the best model in the entire search space is often not the focus or used. Our method achieves or even surpasses the results of training from scratch on the majority of subnetworks, demonstrating its effectiveness in supernetwork training.

Table 3: Effect of our Kshot-Hypernet based on Cifar100. The result of KshotNAS (Su et al., 2021) is produced by us, since they didn't provide source code.

|  | Avg. Acc. | Max. Acc. |
| --- | --- | --- |
| Weight sharing | 53.04% | 65.24% |
| Kshot($N = 12$) | 53.75% | 63.88% |
| Kshot-Hypernet | 60.66% | 68.40% |

**Effectiveness of Kshot-Hypernet:** We compare the performance of direct weight sharing, Kshot-NAS (Su et al., 2021), and our Kshot-Hypernet method on CIFAR-100. As shown in Table 3, our method achieves an improvement of approximately 7% in average accuracy and about 5% in maximum accuracy. These results demonstrate that our approach significantly outperforms other methods in supernetwork training within *TSS*.

Table 4: Effect of our distillation method based on Cifar100.

|  | Avg. Acc. | Max. Acc. |
| --- | --- | --- |
| w/o KD | 58.60% | 67.86% |
| w/ our KD | 60.26% | 69.30% |

**Effectiveness of our distillation method:** To further validate the effectiveness of our distillation strategy, we conducted an ablation study. As shown in Table 4, our distillation method improves both the average and maximum accuracy on CIFAR-100 by approximately 1.5%. This demonstrates that our distillation approach effectively enhances the performance of individual subnetworks and overall improves the supernetwork's performance.

**Effectiveness of Focus-Fair Sampling:** As discussed in Section 3.3, previous sampling methods such as FairNAS tend to result in insufficient training for high-performance subnetworks. To evaluate the effectiveness of our Focus-Fair sampling strategy, we compared the training outcomes of both sampling methods. As shown in Figure 6, our approach effectively addresses the performance gap for top-ranked subnetworks caused by FairNAS sampling.

**Ranking ability:** Although ranking ability is not our primary focus, we also evaluated the ranking performance of our method on NAS-Bench-201. As shown in Table 5, our method achieves a Kendall's Tau value of approximately 70% on both datasets. On ImageNet16-120, the ranking ability remains significantly better than other methods, with a Kendall's Tau value of 63.95%. This indicates that our method surpasses nearly all other approaches in ranking ability, further validating its effectiveness

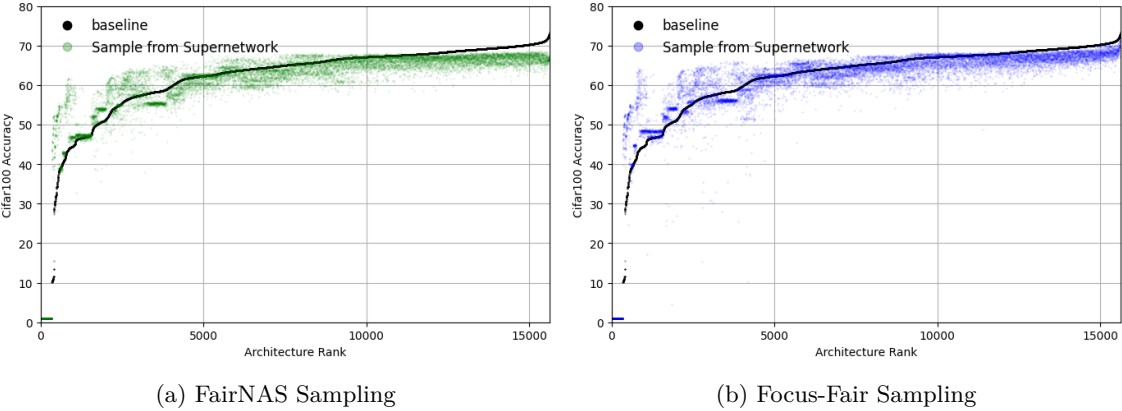

(a) FairNAS Sampling                    (b) Focus-Fair Sampling

Figure 6: Comparison of FairNAS Sampling and Focus-Fair Sampling. The baseline is the result of training from scratch, provided by NAS-Bench-201.

Table 5: Kendall's Tau of different methods on NAS-Bench-201.

| Method | Cifar10 | Cifar100 | Imagenet16 |
|---|---|---|---|
| SPOS Guo et al. (2020) | 55.00% | 56.00% | 54.00% |
| AngleNet Hu et al. (2020) | 57.48% | 60.40% | 54.45% |
| K-shot Su et al. (2021) | 62.64% | 61.22% | 56.33% |
| FewShot(25-supernets) Zhao et al. (2021) | 69.60% | N/A | N/A |
| Ours | 69.42% | 70.18% | 63.95% |

Table 6: Results on ImageNet-1K

| Model | Params | MACs | Epochs | Top-1 Acc. |
|---|---|---|---|---|
| MobileNet-V2-1.0x Sandler et al. (2019) | 3.4M | 0.3B | 500 | 72.0 |
| MobileNet-V3-Large-0.75x Howard et al. (2019) | 4.0M | 0.2B | 500 | 73.3 |
| MNv4-Conv-S Qin et al. (2024) | 3.8M | 0.2B | 9600 | 73.8 |
| iFormer-T Zheng (2025) | 2.9M | 0.5B | 300 | 74.1 |
| FastViT-T8 Vasu et al. (2023a) | 3.6M | 0.7B | 300 | 75.6 |
| **KHyper-S** | **3.9M** | **0.5B** | **225** | **76.0** |
| MobileNet-V3-Large 1.0x Howard et al. (2019) | 5.4M | 0.2B | 500 | 75.2 |
| MobileNet-V2 1.5x Sandler et al. (2019) | 6.8M | 0.7B | 500 | 76.8 |
| MobileOne-S2 Vasu et al. (2023b) | 7.8M | 1.3B | 300 | 77.4 |
| MobileViG-S Munir et al. (2023) | 7.2M | 1.0B | 300 | 78.2 |
| RepViT-M1.0 Wang et al. (2024) | 6.8M | 1.1B | 300 | 78.6 |
| iFormer-S Zheng (2025) | 6.5M | 1.1B | 300 | 78.8 |
| EfficientNet-B1 Tan & Le (2019) | 7.8M | 0.7B | 350 | 79.1 |
| **KHyper-M** | **6.8M** | **1.3B** | **225** | **79.1** |
| MIT-EfficientViT-B1-r224 Cai et al. (2024) | 9.1M | 0.5B | 450 | 79.4 |
| FastViT-S12 Vasu et al. (2023a) | 8.8M | 1.8B | 300 | 79.8 |
| MNv4-Conv-M Qin et al. (2024) | 9.2M | 1.0B | 500 | 79.9 |
| EfficientNet-B2 Tan & Le (2019) | 9.2M | 1.0B | 350 | 80.1 |
| iFormer-M Zheng (2025) | 8.9M | 1.6B | 300 | 80.4 |
| FastViT-SA12 Vasu et al. (2023a) | 10.9M | 1.9B | 300 | 80.6 |
| MNv4-Hybird-M Qin et al. (2024) | 10.5M | 1.2B | 500 | 80.7 |
| **KHyper-L** | **8.7M** | **1.8B** | **225** | **80.7** |

## 4.2 Evaluation on ImageNet-1K

**Dataset:** We also evaluate our method on ImageNet-1K, a large-scale image classification dataset containing 1.28 million training images across 1000 categories.

**Search space:** We use the UIB block from MobileNetV4 (Qin et al., 2024) as our search space, which consists of two depthwise (DW) convolutions and two pointwise (PW) convolutions. Additionally, we also search for model size. The detailed configuration of the search space is detailed in Appendix A.3.

**Supernetwork Training:** We adopt the same Hypernetwork configuration as utilized in NAS-Bench-201. SGD with a momentum of 0.9 and an initial learning rate of 0.1 is employed for optimization, with training conducted over 225 epochs and a batch size of 2048. Cosine learning rate decay is applied throughout the training process. For further details regarding data augmentation and additional training strategies, please refer to Appendix A.1. The training is conducted on 16 Nvidia H100 GPUs. Due to the multiple samplings required for supernetwork training and the additional computation needed by the Hypernetwork to generate weights, the complete training process consumes approximately 1500 GPU hours.

**Result:** Table 6 presents a comparison between our models and state-of-the-art (SOTA) models on ImageNet-1K. The detailed configurations of the architectures discovered by our search are provided in Appendix A.4. Our small model achieves a Top-1 accuracy of 76.0% with 3.9M parameters and 0.5B MACs, outperforming MobileNet-V3 (Howard et al., 2019) by approximately 2.7% at a similar parameter scale. It also surpasses the latest iFormer-T (Zheng, 2025) by 1.9%. The medium model attains a Top-1 accuracy of 79.1% with 6.8M parameters and 1.3B MACs, exceeding RepViT-M1.0 (Wang et al., 2024) and iFormer-S (Zheng, 2025) by approximately 0.5% and 0.3%, respectively, at comparable parameter counts and MACs. Our large model reaches a Top-1 accuracy of 80.7% with 8.7M parameters and 1.8B MACs, slightly surpassing iFormer-M (Zheng, 2025) by 0.3% at a similar parameter count. Compared to MNv4-Conv-M (Qin et al., 2024), although it requires more MACs, it achieves higher accuracy. Overall, our approach achieves competitive accuracy on ImageNet-1K compared to SOTA models, without any additional retraining or fine-tuning.

## 5 Conclusion

In this paper, we propose a novel training paradigm for *TSS*-based supernetwork. After training the supernetwork, it can be directly searched and deployed on the target platform without the need for retraining. We introduce a new distillation and sampling method for *TSS*-NAS, which effectively improves the performance of all architectures in the search space after supernetwork training. Our method transcends the limitations of the MobileNet search space, enabling the training of a supernetwork to be applicable across various platforms, thereby increasing flexibility and efficiency in deployment.

We conducted experiments on NAS-Bench-201, achieving results comparable to training from scratch for most sub-networks. We also performed experiments on ImageNet, achieving 80.5% Top-1 accuracy with 8.6M parameters, surpassing MobileNetV4-M. The results demonstrate the effectiveness of our method in training a supernetwork that can be efficiently searched and deployed on various platforms.

In summary, our approach provides a promising direction for future research in NAS, particularly in the context of topological search spaces. We believe that our method can serve as a foundation for further advancements in NAS, enabling more efficient and effective architecture search and deployment across diverse platforms. We hope that our work will inspire further research in this area and contribute to the development of more efficient and effective NAS methods.

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

# A Appendix

## A.1 Training Details

In this section, we will introduce the training details for each dataset. For all experiments, we set the same hypernetwork hyperparameters $h = 32$ and $N = 64$.

**NAS-Bench-201:** To increase the batch size for the architecture feature and weight coefficient generation network during each update, we adopt the same strategy as GHN3 (Knyazev et al., 2023): using identical input samples on each GPU while training with different subnetworks. We utilize 4 NVIDIA H100 GPUs for training. According to the NAS-Bench-201 setup, 5 subnetworks are sampled per iteration (corresponding to the 5 possible operations), resulting in a batch size of 20 for the architecture feature and weight coefficient generation network in each iteration. Training is performed using SGD with momentum of 0.9 and Nesterov acceleration. Each GPU uses a batch size of 256. The initial learning rate is set to 0.2 and decayed

using a cosine schedule over 300 epochs. For Focus-Fair sampling, the temperature parameter is set to $\tau = 1.5$. We also employ the same warm-up strategy as KshotNAS (Su et al., 2021), where all weight coefficients are set equal during the first 5 epochs. For in-place distillation, we follow the weight decay strategy of BigNAS (Yu et al., 2020a), applying weight decay only to the teacher network, and removing weight decay from all BatchNorm layers and biases. The complete training process is summarized in Algorithm 1. After supernetwork training, we directly evaluate subnetworks sampled from the supernetwork without any retraining. During evaluation, we follow Yu et al. (2018) and use 2048 training samples to recompute the running statistics of the BatchNorm layers. To ensure fairness, our experiments strictly follow the NAS-Bench-201 protocol, with no additional data augmentation.

**ImageNet-1K:** Due to the larger size of ImageNet-1K, we only use DDP for training. The training learning rate is set to 0.1, using SGD with a momentum of 0.9 and a batch size of 2048, with cosine learning rate decay. We also employ the sandwich rule from BigNAS (Yu et al., 2020a), where each sampled architecture undergoes Focus-Fair sampling with a temperature of 1.5. Instead of the warm-up strategy used in NAS-Bench-201, we apply a temperature annealing strategy, where the temperature decreases from 30 to 1 over the first 10 epochs to facilitate smoother optimization. L2 normalization and Dropout are employed for regularization. For data augmentation, we use RandAugment (Cubuk et al., 2019), CutMix (Yun et al., 2019), and Mixup (Zhang et al., 2018), with configurations consistent with MNv4-Conv-L (Qin et al., 2024).

---

**Algorithm 1** Training a Topology Search Space Based Supernetwork

---

**Input:** number of training epochs $E$, warmup epochs $E_w$, training data loader $D$, number of generated weights $N$, weight coefficients $\lambda$, operations pool $O$, number of operation candidates $K$.
**for** $e = 0$ **to** $E - 1$ **do**
    **if** $e < E_w$ **then**
        set warmup temperature of $\lambda_n$; # warmup phase
    **end if**
    train the entire supernetwork and get the soft label $\hat{y}$;
    calculate the loss with true label $y$ and backward pass;
    update weights and $\beta$ with weight decay;
    **for** $k = 0$ **to** $K - 1$ **do**
        randomly sample one architecture from $O$ with weights $softmax(\beta/\tau)$;
        remove the sampled operation from $O$, and its corresponding architecture parameter from $\beta$;
        train the sampled architecture with soft label $\hat{y}$;
    **end for**
    update weights without weight decay;
**end for**

---

## A.2 Ablation study on NAS-Bench-201

**Rank of Decomposition:** As introduced in Section 3.1, the hypernetwork is based on weight decomposition. An important hyperparameter in weight decomposition is the rank. We further investigate its impact on training the hypernetwork for NAS. We conduct experiments with ranks of 8, 16, and 32, and present the results in Table 7. As the rank increases, both the average and maximum performance of the hypernetwork improve, indicating that a larger rank enhances the expressive power of the hypernetwork. However, the impact on Kendall's tau is minimal, suggesting that simply increasing the rank does not significantly improve the training of high-performance subnetworks.

**Number of $N$ for Kshot-Hypernet:** Another important hyperparameter in our Kshot-Hypernet is the number of $N$. Since we represent weights using weight decomposition and $N$ is applied only to one of the decomposition matrices, its impact may differ from that in Kshot-NAS. We analyze this by conducting experiments with $N$ values of 8, 16, 32, and 64, and present the results in Table 8. When $N$ is less than 64, both the average and maximum performance of the hypernetwork improve as $N$ increases. However, the improvement gradually slows down with larger $N$ values. At $N = 64$, both average and maximum performance decrease. This may be due to a lack of sufficient overfitting prevention methods during CIFAR100 training,

Table 7: Impact of rank on Decomposition. We compare the average performance, maximum performance, and Kendall's tau of the hypernetwork with ranks of 8, 16, and 32.

|  | Avg. Acc. | Max. Acc. | Kendall's Tau |
|---|---|---|---|
| *Rank* = 8 | 32.12% | 40.80% | 41.26% |
| *Rank* = 16 | 40.25%(+8.13%) | 53.30%(+12.5%) | 40.94%(-0.32%) |
| *Rank* = 32 | 44.18%(+12.06%) | 57.20%(+16.4%) | 39.13%(-2.13%) |

Table 8: Impact of number of $N$ in Kshot-Hypernet. We compare the average performance, maximum performance, and Kendall's tau of the hypernetwork with $N$ values of 8, 16, 32, and 64.

|  | Avg. Acc. | Max. Acc. | Kendall's Tau |
|---|---|---|---|
| $N = 8$ | 56.71% | 65.22% | 65.19% |
| $N = 16$ | 58.07%(+1.36%) | 65.62%(+0.4%) | 66.14%(+0.95%) |
| $N = 32$ | 59.46%(+2.75%) | 66.88%(+1.66%) | 64.61%(-0.58%) |
| $N = 64$ | 59.21%(+2.5%) | 66.60%(+1.38%) | 63.35%(-1.84%) |

leading to a performance bottleneck for the hypernetwork. The ranking ability of the hypernetwork shows only slight fluctuations as $N$ increases.

### A.3 Search space details

The detailed configuration of the size search space is presented in Table 9. The initial layer is fixed as a $3 \times 3$ convolutional layer with a stride of 2 and an output channel size of 32. Each search block supports four candidate structures: Extra DW, Inverted Bottleneck, ConvNext-Like, and FFN. The SE module is excluded due to its limited hardware efficiency. The classification head adopts the same structure and channel size as MobileNetV3. Upon completion of training, we conduct an evolutionary search with a population size of 100 over 50 iterations. In each iteration, the top 20 subnetworks are selected as parents, 50 subnetworks are generated through mutation, and another 50 are produced via crossover.

Table 9: Size search space details.

| Stage | Channels | Depth | Kernel Sizes |
|---|---|---|---|
| 1 | [16, 24] | [1, 2] | [3, 5] |
| 2 | [24, 32] | [2, 3] | [3, 5] |
| 3 | [40, 48] | [2, 3] | [3, 5] |
| 4 | [80, 88] | [2, 3, 4] | [3, 5] |
| 5 | [112, 128] | [2, 3, 4, 5, 6] | [3, 5] |
| 6 | [192, 216] | [2, 3, 4, 5, 6] | [3, 5] |
| 7 | [320, 352] | [1, 2] | [3, 5] |

### A.4 Model details

Tables 10 to 12 present the detailed parameter settings for the small, medium, and large models. In the early stages, all three models frequently use the ExtraDW, IB, and ConvNext modules. In the final stage, only the FFN module is used for channel fusion, which is consistent with the search results of MobileNetV4 Qin et al. (2024).

Table 10: KHyper-S architecture details

| Input | Block | $K_1$ | $K_2$ | Expand ratio | Output Dim | Stride |
|-------|-------|-------|-------|--------------|------------|--------|
| $272^2 \times 3$ | Conv3 $\times$ 3 | - | - | - | 32 | 2 |
| $136^2 \times 32$ | IB | - | 3 | 1 | 16 | 1 |
| $136^2 \times 16$ | FFN | - | - | 1 | 16 | 1 |
| $136^2 \times 16$ | ExtraDW | 3 | 3 | 6 | 24 | 2 |
| $68^2 \times 24$ | ExtraDW | 5 | 3 | 6 | 24 | 1 |
| $68^2 \times 24$ | ExtraDW | 5 | 3 | 6 | 48 | 2 |
| $34^2 \times 48$ | ExtraDW | 5 | 5 | 6 | 48 | 1 |
| $34^2 \times 48$ | ConvNext | - | 5 | 6 | 48 | 1 |
| $34^2 \times 48$ | ExtraDW | 5 | 3 | 6 | 88 | 2 |
| $17^2 \times 88$ | IB | - | 5 | 6 | 88 | 1 |
| $17^2 \times 88$ | IB | - | 3 | 6 | 112 | 1 |
| $17^2 \times 112$ | ConvNext | 3 | - | 6 | 112 | 1 |
| $17^2 \times 112$ | ExtraDW | 5 | 5 | 6 | 112 | 1 |
| $17^2 \times 120$ | IB | - | 5 | 6 | 192 | 2 |
| $8^2 \times 192$ | ConvNext | 3 | - | 6 | 192 | 1 |
| $8^2 \times 192$ | IB | - | 5 | 6 | 192 | 1 |
| $8^2 \times 192$ | FFN | - | - | 6 | 320 | 1 |
| $8^2 \times 320$ | Conv1 $\times$ 1 | - | - | - | 960 | 1 |
| $8^2 \times 960$ | GlobalAvgPool | - | - | - | 960 | 1 |
| $1^2 \times 960$ | Conv1 $\times$ 1 | - | - | - | 1280 | 1 |
| $1^2 \times 1280$ | Conv1 $\times$ 1 | - | - | - | 1000 | 1 |

Table 11: KHyper-M architecture details

| Input | Block | $K_1$ | $K_2$ | Expand ratio | Output Dim | Stride |
|---|---|---|---|---|---|---|
| $320^2 \times 3$ | Conv3 $\times$ 3 | - | - | - | 32 | 2 |
| $160^2 \times 32$ | FFN | - | - | 1 | 16 | 1 |
| $160^2 \times 16$ | ExtraDW | 5 | 3 | 6 | 32 | 2 |
| $80^2 \times 32$ | ExtraDW | 3 | 3 | 6 | 32 | 1 |
| $80^2 \times 32$ | IB | - | 5 | 6 | 48 | 2 |
| $40^2 \times 48$ | ExtraDW | 5 | 3 | 6 | 48 | 1 |
| $40^2 \times 48$ | ExtraDW | 3 | 3 | 6 | 88 | 2 |
| $20^2 \times 88$ | IB | - | 5 | 6 | 88 | 1 |
| $20^2 \times 88$ | FFN | - | - | 6 | 88 | 1 |
| $20^2 \times 88$ | ExtraDW | 5 | 5 | 6 | 88 | 1 |
| $20^2 \times 88$ | ExtraDW | 5 | 3 | 6 | 128 | 1 |
| $20^2 \times 128$ | ConvNext | 3 | - | 6 | 128 | 1 |
| $20^2 \times 128$ | IB | - | 5 | 6 | 128 | 1 |
| $20^2 \times 128$ | FFN | - | - | 6 | 128 | 1 |
| $20^2 \times 128$ | ExtraDW | 3 | 5 | 6 | 128 | 1 |
| $20^2 \times 128$ | IB | - | 5 | 6 | 216 | 2 |
| $10^2 \times 216$ | ConvNext | 5 | - | 6 | 216 | 1 |
| $10^2 \times 216$ | ExtraDW | 3 | 5 | 6 | 216 | 1 |
| $10^2 \times 216$ | ConvNext | 5 | - | 6 | 216 | 1 |
| $10^2 \times 216$ | FFN | - | - | 6 | 320 | 1 |
| $10^2 \times 320$ | FFN | - | - | 6 | 320 | 1 |
| $10^2 \times 320$ | Conv1 $\times$ 1 | - | - | - | 960 | 1 |
| $10^2 \times 960$ | GlobalAvgPool | - | - | - | 960 | 1 |
| $1^2 \times 960$ | Conv1 $\times$ 1 | - | - | - | 1280 | 1 |
| $1^2 \times 1280$ | Conv1 $\times$ 1 | - | - | - | 1000 | 1 |

Table 12: KHyper-L architecture details

| Input | Block | $K_1$ | $K_2$ | Expand ratio | Output Dim | Stride |
|---|---|---|---|---|---|---|
| $320^2 \times 3$ | Conv3 $\times$ 3 | - | - | - | 32 | 2 |
| $160^2 \times 32$ | IB | - | 5 | 1 | 24 | 1 |
| $160^2 \times 24$ | IB | - | 5 | 1 | 24 | 1 |
| $160^2 \times 24$ | ExtraDW | 5 | 5 | 6 | 32 | 2 |
| $80^2 \times 32$ | ExtraDW | 3 | 5 | 6 | 32 | 1 |
| $80^2 \times 32$ | ExtraDW | 3 | 5 | 6 | 32 | 1 |
| $80^2 \times 32$ | IB | - | 5 | 6 | 48 | 2 |
| $40^2 \times 48$ | IB | - | 5 | 6 | 48 | 1 |
| $40^2 \times 48$ | ExtraDW | 5 | 5 | 6 | 48 | 1 |
| $40^2 \times 48$ | FNN | - | - | 6 | 88 | 2 |
| $20^2 \times 88$ | ExtraDW | 3 | 5 | 6 | 88 | 1 |
| $20^2 \times 88$ | ConvNext | 5 | - | 6 | 88 | 1 |
| $20^2 \times 88$ | ExtraDW | 5 | 5 | 6 | 88 | 1 |
| $20^2 \times 88$ | IB | - | 5 | 6 | 128 | 1 |
| $20^2 \times 128$ | ConvNext | 5 | - | 6 | 128 | 1 |
| $20^2 \times 128$ | ExtraDW | 5 | 5 | 6 | 128 | 1 |
| $20^2 \times 128$ | FFN | - | - | 6 | 128 | 1 |
| $20^2 \times 128$ | IB | - | 5 | 6 | 128 | 1 |
| $20^2 \times 128$ | IB | - | 5 | 6 | 128 | 1 |
| $20^2 \times 128$ | ExtraDW | 5 | 5 | 6 | 216 | 2 |
| $10^2 \times 216$ | ConvNext | 3 | - | 6 | 216 | 1 |
| $10^2 \times 216$ | IB | - | 5 | 6 | 216 | 1 |
| $10^2 \times 216$ | IB | - | 5 | 6 | 216 | 1 |
| $10^2 \times 216$ | IB | - | 5 | 6 | 216 | 1 |
| $10^2 \times 216$ | ConvNext | 5 | - | 6 | 216 | 1 |
| $10^2 \times 216$ | FFN | - | - | 6 | 352 | 1 |
| $10^2 \times 352$ | FFN | - | - | 6 | 352 | 1 |
| $10^2 \times 352$ | Conv1 $\times$ 1 | - | - | - | 960 | 1 |
| $10^2 \times 960$ | GlobalAvgPool | - | - | - | 960 | 1 |
| $1^2 \times 960$ | Conv1 $\times$ 1 | - | - | - | 1280 | 1 |
| $1^2 \times 1280$ | Conv1 $\times$ 1 | - | - | - | 1000 | 1 |

