# OpenReview forum: "Single Train Multi Deploy on Topology Search Spaces using Kshot-Hypernet"
_TMLR — Rejected by TMLR_

### Review · Reviewer_wPpM · 2025-09-05

**Summary Of Contributions:**

## Summary & Contribution
This paper introduces Kshot-Hypernet, a novel neural architecture search (NAS) framework that enables single-train, multi-deploy functionality in Topology Search Spaces (TSS)—a significant departure from prior work that has largely focused on Size Search Spaces (SSS). The key contribution of this paper can be summarized as follows:
- Hypernetwork-based weight generation, improving parameter efficiency;
- K-shot weight dictionary decomposition, enhancing expressiveness;
- In-place distillation using the aggregated supernet as a teacher;
- Focus-Fair sampling, balancing fairness with bias toward stronger subnetworks.
- Strong empirical results, achieving competitive performance on NAS-Bench-201 and ImageNet, with most subnetworks matching or exceeding their from-scratch counterparts.

## Strengths
- Clear Motivation: This paper proposes a critical problem that supernet based NAS methods generally need retraining before deployment, which hinders the application of NAS algorithms. The authors incorporate several methods including K-shot, hypernet and distillation to tackle this problem.
- Sufficient Ablations: This paper reports all hyperparameters of training in appendix and main content. Additionally, this paper conducts ablation studies on distillation and K-shot parameters, which makes its methods more plausible.
## Weaknesses
- Bad Clarity: This paper has many ambiguous words and figures that may mislead the readers. See “Requested Changes” below.
- Missing Explanation on “Multi Deploy” Feature: This paper emphasizes on multi-deployment of the proposed Kshot-Hypernet methods in the title. However, none of the experiment deploy their KHyper-{S,M,L} models to different hardware, and there is not explanation about that.
- Questionable Performance of Kshot-Hypernet: Some data listed in table 1,4 are worse than other works, so they can’t support the effectiveness of Kshot-Hypernet method.
- Lack of Experiments: This paper conducts experiments on NAS-bench-201 & ImageNet-1K, which is an old benchmark. New ones like NAS-bench-301 (a darts-like benchmark) are not considered.
- Missing Comparisons: This paper lacks some critical comparisons like DARTS and SMASH.

**Additional Comments:**

This paper must solve some critical problems and improve overall quality to be accepted by TMLR.

**Audience:**

Yes

**Audience Explanation:**

TMLR’s audience includes researchers in machine learning, particularly those working on automated machine learning (AutoML), neural architecture search, and efficient deep learning. This paper introduces an interesting problem in NAS of “Single Train Multi Deploy”, which may motivate researchers of this field.

**Broader Impact Concerns:**

No ethical or social concerns.

**Claims And Evidence:**

No

**Claims Explanation:**

The paper claims that TSS is better for hardware-aware search, but no hardware latency or energy measurements are provided. This weakens the argument that the method is truly hardware-aware. Therefore, the “multi-deploy” feature described in title is not convincing.

Besides, some data can’t support the effectiveness of the proposed methods. For example, Table 1 shows and Figure 6 shows the Kshot-Hypernet has clear gap with the baseline. Table 4 shows that Kshot-Hypernet achieves lower KT correlation on NAS-bench-201 than FewShot(25-supernets).

**Requested Changes:**

## 1. Writing Problems
1.1 (CRITICAL) I find that Figure 7a is identical to Figure 6. Only one should be left in the paper.

1.2 (CRITICAL) Making the training procedure clearer. In Section 3.3, directly according to your statement and illustration of Figure 5, the hypernet is not involved in the training process. This would confuse readers about the usage of hypernet. Maybe you should draw it in Figure 5.

1.3 In Section 4.1 Table 1 (page 8-9) you use the term “baseline”, but the concept is explained in Figure 7 (page 10). You should state the explanation before using it.

1.4 Add more text explanation on Coefficient Attention Block (CAB) besides from the Figure 4, and explain why CAB instead of traditional attention mechanism.

1.5 In Table 4, the column “ours” should be “Ours” (Capitalize first letter).

## 2. Paper Organization Problems
2.1 (CRITICAL) This paper includes “Multi Deploy” in the title, but there is no hardware-related data in the content. Authors should deploy their algorithm on different devices (e.g. GPU and CPU like Figure 1) and test their performance to support your claim.

2.2 Up to Sept. 2025, TSS based search space like NAS-bench-201 are cited more than 1000 times, so it is investigated a lot. Is it still correct to say “Topology Search Space (TSS) remain unexplored” in abstract? If it’s not true, you can just state that TSS is more important than SSS.

## 3. Experiment Problems
3.1 (CRITICAL) In table 1 and Figure 7, there is a clear gap between your algorithm and baseline. Although you say “maximum accuracy between our method and the baseline is within 2%”, the accuracy variation in NAS-bench-201 is rather small. Most architectures’ accuracy is in a small range. Is the 2% gap small enough to support your conclusion? Maybe you should add comparison to classical works like DARTS or ProxylessNAS to prove this core idea.

3.2 (CRITICAL) Besides params and MACs in Table 5, training epochs and distillation also affect the model performance. Since you implement NAS on MobileNetV4 ImageNet-1k search space with distillation, you should compare distillation version of MobileNetV4 (arxiv: 2404.10518) with similar epochs. You should add more information about distillation and training information on Table 5.

3.3 If you have enough resources, you should compare your methods with DARTS, SMASH and other important NAS literatures.

---

### Review · Reviewer_DbjR · 2025-09-26

**Summary Of Contributions:**

This paper addresses the challenge of neural architecture search in topology search spaces (DAG structures), where traditional “train once, deploy many” approaches often fail due to large structural diversity and weight interference among subnets. The authors propose a new framework that combines hypernetwork-based conditional weights (K-shot Hypernet), a Focus-Fair sampling strategy, and distillation training, enabling a single trained supernet to produce subnets that can be directly deployed. Experiments on NAS-Bench-201 demonstrate that the method achieves average performance comparable to training from scratch, with the best subnet slightly lower but still close, showing the feasibility of train-once deploy-many in topology spaces.

**Additional Comments:**

The main concern is that this paper has already been accepted at the ICML 2024 Workshop (https://icml.cc/virtual/2024/37154
), and the vast majority of the content appears identical. It is unclear why the authors are submitting the same work again to TMLR. Clarification is needed regarding what has been added, revised, or extended compared to the previously published workshop paper, and whether the current submission contains sufficient original contributions to justify a second submission.

[After Rebuttal] I note that the authors have not provided a response to my comments. Given the lack of clarification on these points, I would like to defer the final assessment of the paper to the Area Chair.

**Audience:**

Yes

**Audience Explanation:**

This paper is relevant to TMLR’s audience because it tackles a practical and timely problem in neural architecture search by enabling a supernet to be trained once and deploy multiple subnets efficiently in topology search spaces. Researchers and practitioners working on NAS, supernet training, model compression, and efficient deployment would find the methodology and empirical results valuable. The combination of hypernetwork-based conditional weights, Focus-Fair sampling, and distillation training provides novel insights into managing weight sharing and structural diversity, which are challenges directly aligned with TMLR’s focus on machine learning theory, methodology, and empirical validation.

**Broader Impact Concerns:**

This work focuses on methods for neural architecture search and efficient model deployment, which are primarily technical contributions with no direct ethical or societal risks. It does not involve human subjects, sensitive data, or decisions that could negatively impact individuals. The broader impact is therefore limited, as the methods aim to improve training efficiency and model performance without introducing ethical concerns.

**Claims And Evidence:**

Yes

**Claims Explanation:**

The claims in this paper are well supported by clear empirical evidence. The authors provide comprehensive experiments on NAS-Bench-201 showing that their proposed framework combining hypernetwork-based conditional weights, Focus-Fair sampling, and distillation training allows subnets to be directly deployed with average performance nearly matching that of training from scratch. The methodology is described clearly and the results convincingly demonstrate the feasibility of train once deploy many in topology search spaces. This work is likely to interest TMLR readers, particularly those working on neural architecture search, efficient model deployment, and supernet training, as it addresses a recognized challenge in the NAS community. While the contribution is focused on topology search spaces and validated on a benchmark dataset rather than large-scale real-world models, the approach is novel and provides practical insights that could guide further research. Overall, the paper meets TMLR’s acceptance criteria as claims are supported, the writing is clear, and the work presents a meaningful original advancement rather than a mere reproduction.

**Requested Changes:**

The submission is generally clear and well-executed, but a few adjustments could strengthen the work.

- Providing additional experiments on larger or more realistic topology search spaces beyond NAS-Bench-201 would help demonstrate broader applicability and would strengthen the work, though it is not critical for acceptance.

- Including ablation studies isolating the contributions of hypernetwork conditioning, Focus-Fair sampling, and distillation would clarify the importance of each component and improve reproducibility; this would strengthen the work but is not critical.

- More implementation details such as hyperparameter choices and training stability considerations could aid replication and understanding; this is recommended but not critical.

Overall, these adjustments would enhance the clarity and impact of the submission without being essential for its acceptance.

---

### Review · Reviewer_7MAB · 2025-12-19

**Summary Of Contributions:**

The authors introduce a novel supernetwork formulation that synthesizes Hypernetworks with K-shot NAS. This approach theoretically enables a single set of parameters to dynamically adapt to diverse topological contexts. The authors also propose Focus-Fair Sampling and extend the concept of in-place distillation, popularized in SSS by BigNAS, to the topological domain.

#### **Strengths:**

The central achievement of this work is empirical evidence that sub-networks sampled from a TSS supernetwork can match or exceed the performance of stand-alone training. The fusion of Hypernetworks and K-shot dictionaries is a sophisticated engineering solution to the expressivity bottleneck. Standard Hypernetworks struggle to generate high-fidelity weights for deep CNNs directly.

#### **Weaknesses:**

1. Outdated Comparative Baselines. The framing of sota performance is fragile. The manuscript compares primarily against MobileNetV3 (2019), MobileOne (2022), and EfficientNet (2019). It largely ignores recent efficient networks released in late 2024 and early 2025, specifically MobileNetV4 , RepViT , and iFormer.
2. Lack of Training Cost Transparency. A primary motivation for "Single Train" methods is efficiency. However, training a Hypernetwork-based supernetwork is computationally expensive due to the overhead of weight generation and additional gradient paths. The manuscript lacks a detailed accounting of GPU-hours required for the supernetwork training versus standard methods.

**Audience:**

Yes

**Audience Explanation:**

The TMLR audience, which includes researchers in AutoML and hardware-aware learning, will value a method that makes TSS viable again.

**Broader Impact Concerns:**

Not available.

**Claims And Evidence:**

Yes

**Claims Explanation:**

The claim "Sub-networks match or exceed training from scratch." is accurate and well-supported. The authors provide compelling evidence from NAS-Bench-201, a benchmark specifically designed to test this property because it provides ground-truth "train-from-scratch" data for every architecture in the search space.

**Requested Changes:**

To ensure the manuscript meets the high standards of TMLR, I recommend a Major Revision. The following detailed modifications are necessary to strengthen the paper’s impact, accuracy, and completeness.

1. Comparison with Modern SOTA (Critical). The current results table (Table 5) compares against outdated baselines (MobileNetV3, MobileOne). It omits the true competitors from 2024/2025.
2. Analysis of Training Costs. The paper claims "high efficiency" but does not quantify the cost of the "Single Train" phase.
3. Latency Analysis on Real Hardware. The paper reports MACs and Parameters. However, recent works (MobileNetV4, RepViT) emphasize Latency on real hardware (e.g., iPhone 12/13 latency in ms) because MACs do not correlate perfectly with speed.

---

> ### Author Response · Authors · 2025-12-29
> **Official Comment by Authors**
>
> We thank the reviewer for the careful evaluation and constructive feedback. We address the concerns point by point below.
>
> ---
>
> ### 1. Comparison with Modern SOTA Architectures
>
> We would like to clarify that **comparisons with recent efficient architectures are already included in the current manuscript**.
> Specifically, **RepViT, MobileNetV4, and iFormer** are reported in **Table 5** and discussed in **Section 4.2**.
>
> We apologize if this was not sufficiently emphasized in the presentation. In the revised version, we will improve the clarity and visibility of these results in the main text.
>
> ---
>
> ### 2. Training Cost Transparency
>
> We agree with the reviewer that training cost is an important aspect of *Single-Train* approaches.
>
> While the current submission focuses on accuracy transferability and deployment-time efficiency, we acknowledge that explicit reporting of training cost would improve transparency. In the revision, we will add a detailed accounting of training cost in **Section 4.2**.
>
> ---
>
> ### 3. Latency on Real Hardware
>
> We agree that real-device latency is an important metric. However, we are unable to add new real-hardware latency measurements within the current revision cycle.
>
> We emphasize that our contribution is orthogonal to hardware-specific latency modeling. Our work primarily validates the feasibility of TSS-based approaches in achieving deployment-level accuracy for subnets obtained from a single training session and multiple searches.
>
> In the revised manuscript, we will clarify this scope explicitly.
>
> ---
> We thank the reviewer again for the insightful comments.

---

### Decision · Action_Editor_6jHE · 2026-02-24

**Recommendation:** Reject

**Audience:**

Yes

**Audience Explanation:**

All reviewers unanimously agree that TMLR's audience would find this work interesting. The paper addresses a timely and practical problem by extending "single train, multi-deploy" functionality to Topology Search Spaces, a significant departure from previous work focused on Size Search Spaces. Researchers and practitioners in AutoML, NAS, and efficient model deployment would value the methodology and insights into managing weight sharing and structural diversity.

**Claims And Evidence:**

No

**Claims Explanation:**

The evidence is only partially convincing as critical gaps remain regarding the paper's core claims. While reviewers acknowledge that empirical results on NAS-Bench-201 support the claim that sub-networks can match or exceed train-from-scratch performance , Reviewer wPpM notes that the absence of real hardware latency or energy measurements fundamentally undermines the central multi-deploy claim. Furthermore, efficiency of the Single Train phase was initially questioned due to a lack of transparency regarding GPU hour costs for training the hypernetwork. Consequently, while technical accuracy in specific benchmarks is present, the broader functional claims lack sufficient empirical validation.

**Resubmission Of Major Revision:**

The authors may consider submitting a major revision at a later time.